# Experiences of oncology researchers in the Veterans Health Administration during the COVID-19 pandemic

Daniel J. Becker[1,2], Kenneth Csehak[3]*, Alexander M. Barbaro[3], Stefanie D. Roman[2], Stacy Loeb[4,5], Danil V. Makarov[4,5], Scott Sherman[5,6], Sahnah Lim[5]

1 Department of Hematology and Medical Oncology, Perlmutter Cancer Center, NYU Langone Medical Center, New York, New York, United States of America, 2 Department of Hematology and Medical Oncology, VA New York Harbor Healthcare System, Manhattan Campus, New York, New York, United States of America, 3 Department of Hematology and Medical Oncology, NYU Grossman School of Medicine, New York, New York, United States of America, 4 Department of Urology, NYU Grossman School of Medicine, New York, New York, United States of America, 5 Department of Population Health, NYU Grossman School of Medicine, New York, New York, United States of America, 6 Department of Medicine, NYU Grossman School of Medicine, New York, New York, United States of America

* Kenneth.Csehak@nyulangone.org

**Data Availability Statement:** All collected data are within the manuscript.

**Funding:** The authors received no specific funding for this work.

## Abstract

The Veterans Health Administration is chartered "to serve as the primary backup for any health care services needed. . .in the event of war or national emergency" according to a 1982 Congressional Act. This mission was invoked during the COVID-19 pandemic to divert clinical and research resources. We used an electronic mixed-methods questionnaire constructed using the Theoretical Domains Framework (TDF) and the Capability, Opportunity, and Motivation (COM-B) model for behavior change to study the effects of the pandemic on VHA researchers. The questionnaire was distributed electronically to 118 cancer researchers participating in national VHA collaborations. The questionnaire received 42 responses (36%). Only 36% did not feel that their research focus changed during the pandemic. Only 26% reported prior experience with infectious disease research, and 74% agreed that they gained new research skills. When asked to describe helpful support structures, 29% mentioned local supervisors, mentors, and research staff, 15% cited larger VHA organizations and 18% mentioned remote work. Lack of timely communication and remote work, particularly for individuals with caregiving responsibilities, were limiting factors. Fewer than half felt professionally rewarded for pursuing research related to COVID. This study demonstrated the tremendous effects of the COVID-19 pandemic on research activities of VHA investigators. We identified perceptions of insufficient recognition and lack of professional advancement related to pandemic-era research, yet most reported gaining new research skills. Individualizing the structure of remote work and ensuring clear and timely team communication represent high yield areas for improvement.

## Introduction

The emergence of COVID-19 and its worldwide spread necessitated an unprecedented shift of resources and energy towards combating the pandemic. On January 21, 2020 the Centers for

**Competing interests:** The authors have declared that no competing interests exist.

Disease Control and Prevention (CDC) confirmed the first case of the novel coronavirus in Washington state, quickly followed by the CDC's confirmation of the first case of person-to-person spread on January 30, 2020 [1, 2]. As confirmed COVID-19 cases continued to increase and reached over 34,000 cases per day in early April, the CDC announced $186 million in funding to bolster the country's pandemic response [3, 4]. By this time, hospital bed capacity was decreasing due to the increase in COVID-19 patients and communities across the country were beginning to feel the strain on the healthcare system [5].

As the largest integrated healthcare system in the United States, the Veterans Health Administration (VHA) is chartered "to serve as the primary backup for any health care services needed by the Department of Defense in the event of war or national emergency," according to a 1982 Congressional Act [6]. This rarely invoked function, referred to as the VHA's "Fourth Mission," was enacted on April 14, 2020, during the height of the first wave of the COVID-19 pandemic, to bolster significant clinical and research resources, and "grew to the greatest scale and scope in VA history" [7]. The VA response included caring for non-veteran patients, accepting mission assignments from the Federal Emergency Management Agency (FEMA), and distributing personal protective equipment (PPE) and other medical supplies to state, local, and community organizations, all while providing undisrupted care to Veterans [6, 8]. In addition to the drastic changes in the delivery of clinical care, the conduct of research was similarly redirected towards understanding the epidemiology, pathogenesis, and management of this novel disease; ultimately nearly 70 VA medical centers across the country were involved in one or more COVID-19 related clinical trials [6, 7].

The COVID-19 pandemic resulted in dramatic changes in the activities of clinicians, investigators, and research staff. In April 2020 the American Cancer Society surveyed all of its funded researchers with 93% of the 487 respondents indicating the COVID-19 pandemic was having a moderate or high impact on their research or training activities [8]. Despite the pivotal role the restructuring of VA clinical and research efforts played in the management of COVID-19, there have been few attempts to directly examine the effect this redistribution had on cancer researchers' experiences during the pandemic.

We surveyed cancer research faculty and staff during a 3-month period starting in late October 2021 when COVID-19 cases were slowly increasing but VA activities had mostly returned to pre-pandemic levels, thus allowing respondents the opportunity to reflect upon their experiences during the height on the pandemic. In our survey, we explored the subjective experience of researchers as their duties and responsibilities shifted in response to COVID-19. This assessment provides novel insight into the challenges faced by VHA cancer researchers and how the COVID-19 pandemic affected their work. These findings may guide future inquiry and contribute to a framework response in the event of inevitable emergencies in the future.

## Methods

We constructed a mixed methods, anonymous electronic questionnaire consisting of 6 demographic questions, 33 statements with Likert scale responses, and 8 free response questions (S1 File). The project and questionnaire were approved by New York Harbor VA Health Care System Research and Development Committee (IRBNet ID 1624189–2). Consent was waived for this study as the research involved no more than minimal risk of harm to participants and data was collected anonymously. The survey was created and housed on the Redcap platform with links distributed via secure email to 118 cancer researchers within the VHA including research faculty, post doctorates, and research assistants. These researchers were identified by via two separate research contacts lists, representing participants in national VA lung and prostate

**Table 1. Theoretical domain framework with corresponding COM-B component.**

| COM-B | Capability | Opportunity | Motivation |
|---|---|---|---|
| TDF | Skills | Goals | Belief about consequences |
| | Nature of behavior | Social influences | Emotion |
| | Knowledge | Environmental context and resources | Beliefs about capabilities |
| | Memory, attention, and decision processes | | Reinforcement |
| | | | Intentions |
| | Behavior regulation | | |

cancer consortia, to ensure that respondents were actively involved in oncology related research. Response items consisted of Likert scale questions, i.e., statements regarding researchers' experiences during the COVID-19 pandemic followed by a prompt to select the level of agreement from "strongly disagree" through "strongly agree". Optional, open-ended questions were also included which participants could respond to with free text. We collected respondents' demographic information in addition to the questionnaire responses.

The questionnaire, based on the related conceptual models of the Theoretical Domains Framework (TDF) and Capability-Opportunity-Motivation (COM-B) Behavior Change Wheel, was designed to explore researchers' experience with changes in the nature and environment of their work during the COVID-19 pandemic [9, 10]. The TDF categorizes health related behavioral sources into 14 distinct domains. The COM-B model posits that behavioral change is a product of capability, opportunity, and motivation and aggregates factors influencing these sources from the TDF domains. Table 1 illustrates the TDF domains assessed in this survey and their corresponding COM-B categories.

Likert scale questions were categorized in terms of their relevance to individual TDF domains and responses are quantitatively reported. Open ended responses were coded and analyzed by 4 independent researchers and responses were coded into themes using a content analysis approach. These themes were aggregated and similarly categorized into the appropriate TDF domain. The combination of Likert responses and themes extracted from open-ended questions were used to present multifaceted exploration of the factors influencing behavior and behavior change related to research during the COVID-19 pandemic within each domain.

## Results

### Respondent characteristics

An electronic questionnaire was sent to 118 individuals within the VHA (S1 File). A total of 42 responses were received, for a completion rate of 36%. Demographic data including age, advanced degree type, and geographical distribution was collected (Table 2). Questionnaire items were grouped by domain using the Theoretical Domains Framework and mapped to corresponding COM-B component for analysis. The distribution of selected key Likert scale responses is presented in Fig 1 and summarized in Table 3.

### Capability

Nearly all respondents, 41 of 42 (98%) strongly agreed or agreed that they are often able to adapt to difficult situations and 30 of 42 (71%) respondents strongly agreed or agreed that they had a good baseline knowledge of research methods applicable to COVID-19. However, implementing these research methods was not without challenges, with 28 of 42 (67%) individuals agreeing that they found research challenging during the COVID-19 pandemic. Only 20 of 42

**Table 2. Respondent demographics.**

| Age Range | Number of respondents, (%) |
|---|---|
| 20–29 | 6 (14%) |
| 30–39 | 10 (24%) |
| 40–49 | 15 (36%) |
| 50–59 | 9 (21%) |
| 60–69 | 2 (5%) |
| 70 and older | 0 (0%) |
| **Gender** | |
| Male | 10 (24%) |
| Female | 30 (71%) |
| Prefer not to answer | 2 (5%) |
| **Region** | |
| West (CA, CO, UT, WA) | 13 (31%) |
| Midwest (IL, MN, MI) | 5 (12%) |
| South (MD, NC, TX, VA) | 6 (14%) |
| Northeast (MA, NY, PA) | 18 (43%) |
| **Highest profession degree** | |
| BA, BS, BSN | 9 (21%) |
| MS, MPA, MPH, MSN, MBA | 12 (29%) |
| MD, MD/PhD, MBSS | 14 (33%) |
| PhD | 6 (14%) |
| PharmD | 1 (2%) |
| **Leadership role held** | |
| Yes | 12 (29%) |
| No | 30 (71%) |

(48%) strongly agreed or agreed that they were able to focus on their active research during the pandemic. When respondents were asked how they think the COVID-19 pandemic will change the utilization and distribution of research resources in the future, 7 of 39 responses were categorized as describing how the pandemic will generally improve research: "more allocation [of] resources for research," "perhaps the importance of clinical research that has been highlighted during the pandemic will increase the resources and speed of clinical research moving forward." Five thought there will be an increase in infectious disease research: "Increased focus on emerging infectious disease threats," "we will be better prepared to address potential future pandemics." Five respondents were concerned that the pandemic will lead to a decrease in non-COVID related research funding: "it took too many resources away from cancer research and there will be an impact from this for years to come," "pressing public health issues will take priority in funding over more chronic less communicable diseases such as cancer." Eleven respondents were simply not sure on possible future effects on resource distribution. The distribution of additional Likert scale items within the TDF domains comprising Capability are seen in Table 3 and Fig 1.

## Opportunity

Most respondents, 29 of 41 (71%), strongly agreed or agreed that their institution had appropriate resources to support COVID-19 research. Both helpful support structures and ways which support felt lacking were examined. In qualitative responses describing helpful support structures, the categories of individuals, larger VA organizations, and support processes

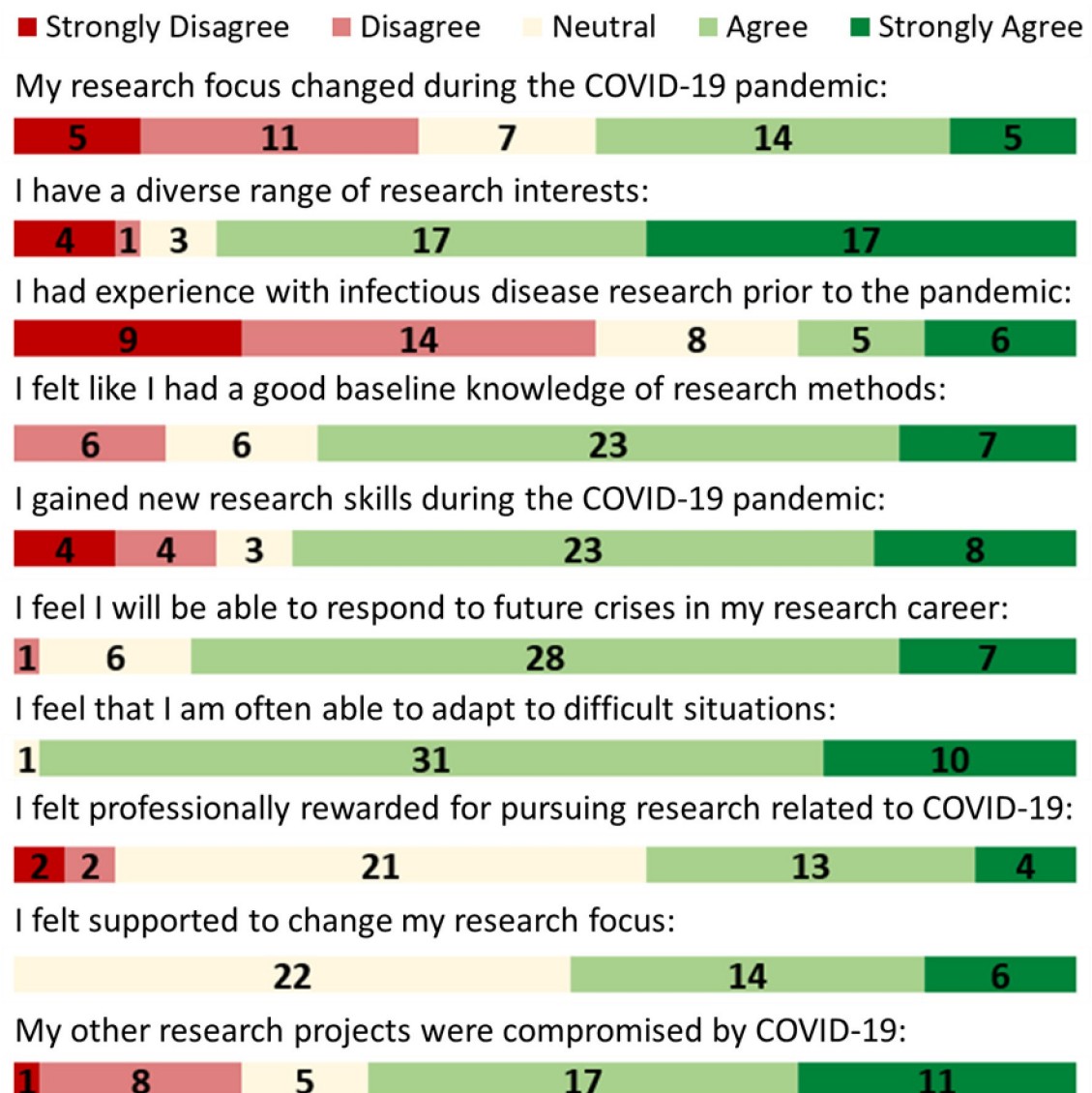

**Fig 1. Response distribution for selected Likert scale items.**

became apparent. Eleven of 39 responses mentioned various individuals as support structures including "colleagues," "coworkers [and] fellow coordinators all helped each other navigate the new rules, regulations, and precautions," and "supervisor support." Larger VA wide organizations such as the Office of Research Development, Prostate Cancer Foundation, and other shared VA resources were also a source of support. Processes such as remote work and telehealth were a source of support, mentioned by 8 respondents, with these processes allowing "limit[ed] exposure to self and patients during peak and respected me as a person and an employee during a universally difficult time for everyone." Regarding ways that support felt lacking, the lack of timely communication from supervisors including about both policies for "opening up research or imposing restrictions" and information about "infection rates" was mentioned. Systems processes such as IRB issues were also mentioned: "too difficult to get IRB

**Table 3. Distribution of responses for Likert scale items within each TDF domain and correlated factors of the COM-B model of behavior change.**

| COM-B | TDF Domain | Statement | Likert scale response, *number (%)* | | | | |
|---|---|---|---|---|---|---|---|
| | | | Strongly Disagree | Disagree | Neutral | Agree | Strongly Agree |
| Capability | Knowledge | I believe I have gained a better understanding about public health during the COVID-19 pandemic | 0 (0%) | 0 (0%) | 3 (7%) | 28 (67%) | 11 (26%) |
| | | I felt like I had a good baseline knowledge of research methods applicable to COVID-19 research | 0 (0%) | 6 (14%) | 6 (14%) | 23 (55%) | 7 (17%) |
| | Skills | I gained new research skills during the COVID-19 pandemic | 4 (10%) | 4 (10%) | 3 (7%) | 23 (55%) | 8 (19%) |
| | | I had experience with infectious disease research prior to the COVID-19 pandemic | 9 (21%) | 14 (33%) | 8 (19%) | 5 (12%) | 6 (14%) |
| | Memory, Attention, Decision Processes | I feel that I was able to focus on my active research during the COVID-19 pandemic | 3 (7%) | 11 (26%) | 8 (19%) | 16 (38%) | 4 (10%) |
| | Nature of behavior | I feel that I am often able to adapt to difficult situations | 0 (0%) | 0 (0%) | 1 (2%) | 31 (74%) | 10 (24%) |
| | | I found research difficult during the COVID-19 pandemic challenging | 0 (0%) | 6 (14%) | 8 (19%) | 15 (36%) | 13 (31%) |
| | | I felt personal responsibility to research COVID-19 | 0 (0%) | 7 (17%) | 10 (24%) | 16 (38%) | 9 (21%) |
| | Behavioral Regulation | I feel I will be able to respond to future crises in my research career | 0 (0%) | 1 (2%) | 6 (14%) | 28 (67%) | 7 (17%) |
| Opportunity | Social Influences | I felt influenced by my peers to change my research focus to a COVID-19 related topic during the pandemic | 6 (14%) | 18 (43%) | 7 (17%) | 7 (17%) | 4 (10%) |
| | | I felt influenced by the public to change my research focus during the COVID-19 pandemic | 7 (17%) | 14 (33%) | 10 (24%) | 10 (24%) | 1 (2%) |
| | | I felt influenced by the VA to change my research focus during the COVID-19 pandemic | 5 (12%) | 13 (31%) | 11 (26%) | 10 (24%) | 3 (7%) |
| | | Personal circumstances were influential in my work during the COVID-19 pandemic | 2 (5%) | 11 (26%) | 11 (26%) | 12 (29%) | 6 (14%) |
| | Environmental Context and Resources | Prior to the COVID-19 pandemic my institution had active investigators focused on infectious disease research | 1 (2%) | 1 (2%) | 14 (33%) | 17 (40%) | 9 (21%) |
| | | My institution has appropriate resources to support COVID-19 research | 0 (0%) | 1 (2%) | 11 (27%) | 21 (51%) | 8 (20%) |
| | | Clinical responsibilities during the COVID-19 pandemic interfered with my research project | 5 (12%) | 12 (29%) | 13 (31%) | 6 (14%) | 6 (14%) |

(*Continued*)

**Table 3.** (Continued)

| COM-B | TDF Domain | Statement | Likert scale response, *number (%)* | | | | |
|---|---|---|---|---|---|---|---|
| | | | **Strongly Disagree** | **Disagree** | **Neutral** | **Agree** | **Strongly Agree** |
| Motivation | Reinforcement | I felt personally rewarded for pursuing research related to COVID-19 | 3 (7%) | 1 (2%) | 20 (48%) | 12 (29%) | 6 (14%) |
| | | I felt personally penalized for pursuing research related to COVID-19 | 9 (21%) | 14 (33%) | 14 (33%) | 3 (7%) | 2 (5%) |
| | | I felt professionally rewarded for pursuing research related to COVID-19 | 2 (5%) | 2 (5%) | 21 (50%) | 13 (31%) | 4 (10%) |
| | | I felt professionally penalized for pursuing research related to COVID-19 | 7 (17%) | 16 (38%) | 16 (38%) | 1 (2%) | 2 (5%) |
| | Emotion | Emotions motivated me in my COVID-19 research | 4 (10%) | 6 (14%) | 14 (33%) | 14 (33%) | 4 (10%) |
| | | Emotions limited me in my COVID-19 research | 6 (14%) | 19 (45%) | 12 (29%) | 4 (10%) | 1 (2%) |
| | Professional Role and Identity | My research focus changed during the COVID-19 pandemic | 5 (12%) | 11 (26%) | 7 (17%) | 14 (33%) | 5 (12%) |
| | | I have a diverse range of research interests | 4 (10%) | 1 (2%) | 3 (7%) | 17 (40%) | 17 (40%) |
| | Beliefs about capabilities | I believe I am able to make a meaningful contribution to improving the COVID-19 pandemic | 0 (0%) | 3 (7%) | 14 (33%) | 20 (48%) | 5 (12%) |
| | Intentions | I have a plan to return or have already returned to my prior research focus | 0 (0%) | 0 (0%) | 8 (19%) | 20 (48%) | 14 (33%) |
| | | I intend to incorporate features of my COVID-19 related research into my research career | 2 (5%) | 5 (12%) | 18 (43%) | 13 (31%) | 4 (10%) |
| | Goals | I believe redistribution of research resources to COVID-19 related research will have an impact on cancer research | 2 (5%) | 0 (0%) | 18 (43%) | 15 (36%) | 7 (17%) |
| | | I achieved my research goals during the COVID-19 pandemic | 0 (0%) | 7 (17%) | 12 (29%) | 18 (43%) | 5 (12%) |
| | Belief about Consequences | I felt supported to change my research focus during the COVID-19 pandemic | 0 (0%) | 0 (0%) | 22 (52%) | 14 (33%) | 6 (14%) |
| | | I felt personal health risk about performing COVID-19 related research | 7 (17%) | 8 (19%) | 13 (31%) | 11 (26%) | 3 (7%) |
| | | I think my research during the COVID-19 pandemic makes me more competitive for future research positions | 3 (7%) | 0 (0%) | 17 (40%) | 14 (33%) | 8 (19%) |
| | | My other research projects were compromised by COVID-19 | 1 (2%) | 8 (19%) | 5 (12%) | 17 (40%) | 11 (26%) |

amendments for VVC [VA Video Connect] consents;" "delayed opening of IRB for projects during the pandemic." Limited in person staff and the negative affect it had on communication was also noted. When asked about how their research resources or support were affected by the COVID-19 pandemic, there was an overall belief that the pandemic had a negative impact on research resources. Twenty of 36 total responses mentioned negative impacts of the pandemic on research resources or support, namely through "reduced staffing," feelings of "being stretched too thin," the problems of either remote work or telehealth or simply logistical challenges like "shipping study labs" due to delays from shipping companies. Five respondents said the pandemic had a positive effect on research or support resources, with one saying, "if anything, I received additional support to do my job safely." Six respondents did not notice a change in research resources or support during the pandemic.

Various social factors and personal circumstances were examined as possible contributing factors that led to individuals changing their research during the COVID-19 pandemic (Table 3). When asked to describe how personal circumstances limited or facilitated research

during the COVID-19 pandemic, the increased reliance on remote work was seen as a positive by some, mentioned by 7 respondents as a facilitator of research. One respondent described how remote work allowed for "increased productivity by reducing commuting time and allowing me to build a wider network remotely." However, remote work was also seen as a drawback for some, particularly those with caregiving responsibilities. Personal or family circumstances including childcare was mentioned as a limiting factor by 10 respondents, with one individual describing how their "young children were home from school for 11 months, therefore [they] had to add the roles of teacher and babysitter to [their] already full schedule" and another who "juggled two positions while also juggling two children under 4 without childcare for the spring/summer of 2020." There were also significant anxieties about health, both personally, for colleagues, and for family members mentioned by 5 respondents.

## Motivation

Our study population of cancer researchers had a diverse range of research interests, and approximately half of respondents, 19 of 42 (45%) strongly agreed or agreed that their research focus changed during the COVID-19 pandemic. A majority, 28 of 42 (67%), strongly agreed or agreed that their other research projects were compromised by COVID-19. Despite significant challenges during the pandemic, 23 of 42 (55%) strongly agreed or agreed that they achieved their research goals.

For most respondents (20 of 37 submitted responses), the pandemic did not change, or served to reaffirm their belief in the importance of research. Representative responses include: "I believe in research strongly, before and during the pandemic," "COVID-19 pandemic dramatically changed my belief about the importance of research in [a] positive way," "made it stronger and showed how important research really is." One respondent described how "COVID laid bare the inefficiencies of VA research." Three respondents described how the pandemic highlighted the rapid change of research: "It only changed in that I saw processes that were mandated historically for research suddenly became fluid. Exceptions to consent methods were made, timing of approvals for research escalated. It was all pretty amazing." Two respondents mentioned remote work in a positive light: "So great that remote meetings are now acceptable so that more people can be engaged and participate from other institutions."

The vast majority of participants, 34 of 42 (81%) have a plan or have already returned to their prior research focus. Seventeen of 42 (40%) respondents plan to incorporate some features of their COVID-19 related research into their future research.

## Discussion

In this study, we used the Theoretical Domains Framework (TDF) and COM-B model, represented in Fig 2, to understand behavioral change among VA cancer researchers during the COVID-19 pandemic. Our findings reveal insights into the domains of behavioral change that influenced these researchers and provide suggestions for future interventions.

Capability played a significant role in the researchers' ability to adapt to the changing landscape inflicted by the pandemic. For the most part, this domain was one in which respondents felt that they were well-equipped to handle the challenges they faced. Despite the difficulties imposed by the extreme external circumstances, respondents rated their baseline skills and experience highly. Furthermore, research professionals felt that the redeployment provided an opportunity for researchers to gather new skills and increase their understanding of public health, and the majority of respondents felt they would be prepared to respond to a future crisis. The VA's strategic plan emphasizes "foster[ing] a culture of continuous learning, including

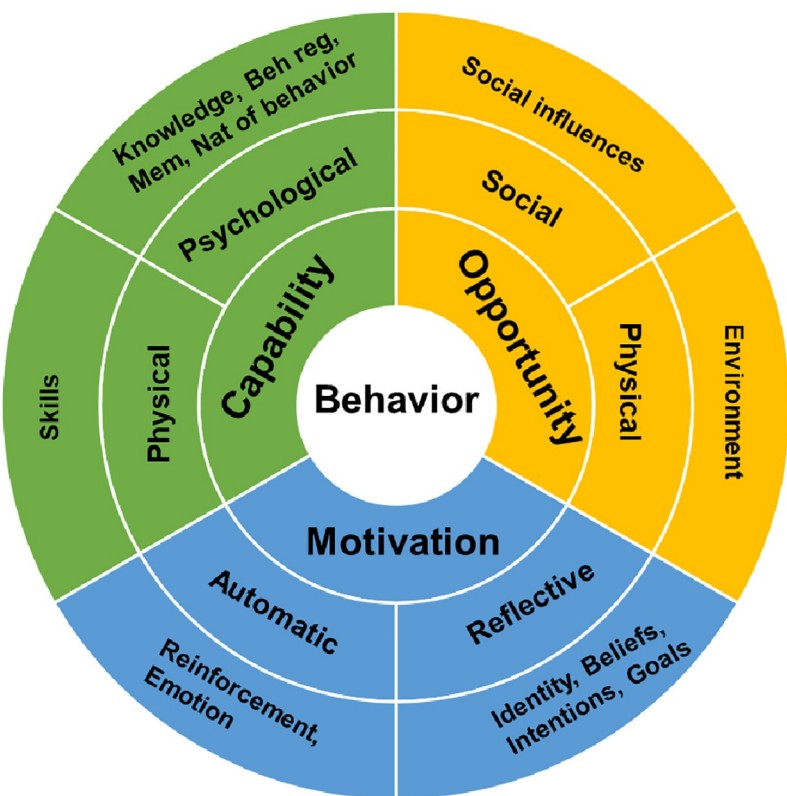

**Fig 2. Representation of the COM-B model for behavior change.** COM-B is a framework for describing the factors of behavior change based on capability, opportunity, and motivations (inner ring). Each factor is divided into subcategories (middle ring) that correspond to the domains of the Theoretical Domains Framework (outer ring). This model was developed by Susan Michie and colleagues [11]. Beh reg: behavioral regulation; Mem: memory; Nat of behavior: nature of behavior.

coaching, mentoring and on-the-job learning" and this approach has enjoyed success in terms of researchers' own confidence in their abilities [12]. Impairments to capability, on the other hand, resulted from the cognitive and attentional burdens imposed by the pandemic. The ways in which these were alleviated or exacerbated by institutional policy are discussed in further depth within the Opportunity and Motivation domains.

Opportunity, in the context of the COM-B model, refers to the external factors that render a change in behavior possible or impossible. Our findings suggest that workplace flexibility and institutional support were key factors influencing the opportunity to change behavior. With regard to workplace flexibility, perhaps the clearest example of this on the clinical side has been the adoption of telehealth services to augment in-office care, and authors have extensively reviewed ways in which these changes were implemented and their effects [13]. Similar, ad-hoc changes were made in the arena of research, such as the movement to remote meetings. Nonetheless, specifics of how institutions catered to the changing needs of research staff during the pandemic this area would benefit from closer scrutiny. While our study helps to define the experience of research personnel during the pandemic, further retrospective review that focuses on concrete measures adopted by institutions are needed. With regard to institutional support, the overwhelming majority of respondents did feel that their institution had resources to support COVID-19 research. This aligns with the fact that, in spite of the novel funding needs presented by the pandemic, the amount of funds allocated to cancer

research according to the NIH still increased from \$6.5 billion to \$7 billion from 2019 to 2020 [14]. Conversely, many researchers felt that staffing issues, particularly redeployment to clinical roles impeded the ability to conduct research. This is well in line with the experience of healthcare workers throughout the field, who noted such issues as reduction in training opportunities and time to review literature [15, 16]. As systems evolve to be more resilient to future disruptions, adequate staffing to prevent the need for redeployment is likely to be a key element.

Motivation was another crucial aspect of behavioral change. Researchers reported possessing strong internal drives, with a robust belief in the importance of their work and flexible goals. However, the majority felt ambivalent about whether they were supported in changing their research focus. Taken together these results seem to suggest that researchers' motivation was mainly derived from their own sense of duty and value in their work, rather than external incentivization. The exception is the reported gain of new research skills during the pandemic. Indeed, other studies have suggested that the concept of "benefit-finding", a process mainly described in cancer patients by which an individual identifies positive growth experiences resulting from trauma, can be applied to the experience of COVID-19 [17]. Overall these results suggest that while individuals are able to draw on reserves of internal motivations, institutions should support this feeling by incentivizing desired behaviors, and helping guide employees towards an understanding of the new skills and capabilities that may have been gained through a difficult experience.

There are some limitations to our study, such as the low response rate, which might affect the generalizability of the findings. Additionally, a more in-depth comparison with similar studies focusing on the institutional, rather than individual, side would provide valuable context for our results. The strengths of this study include the use of a well-established framework for understanding behavioral change and the firsthand accounts of researchers' experiences.

## Conclusion

In conclusion, the VHA's "Fourth Mission" was tested during the COVID-19 pandemic, prompting significant changes in researchers' responsibilities and focus. Our study provides a first glance into these experiences and points towards future lines of inquiry and intervention. Further research is needed to understand the strengths and weaknesses of the response, as well as to develop strategies for improving the system's preparedness for future disruptions.

## Supporting information

**S1 File. Electronic questionnaire distributed to participants.**
(PDF)

## Author Contributions

**Conceptualization:** Daniel J. Becker, Stefanie D. Roman, Stacy Loeb, Danil V. Makarov, Scott Sherman, Sahnah Lim.

**Data curation:** Daniel J. Becker, Kenneth Csehak, Alexander M. Barbaro, Stefanie D. Roman.

**Formal analysis:** Daniel J. Becker, Kenneth Csehak, Alexander M. Barbaro, Stefanie D. Roman.

**Investigation:** Daniel J. Becker, Kenneth Csehak, Alexander M. Barbaro, Stefanie D. Roman.

**Methodology:** Daniel J. Becker, Kenneth Csehak, Alexander M. Barbaro, Stefanie D. Roman, Stacy Loeb, Danil V. Makarov, Scott Sherman, Sahnah Lim.

**Project administration:** Daniel J. Becker, Kenneth Csehak, Alexander M. Barbaro, Stefanie D. Roman.

**Resources:** Daniel J. Becker, Kenneth Csehak, Alexander M. Barbaro, Stefanie D. Roman.

**Software:** Daniel J. Becker, Kenneth Csehak, Alexander M. Barbaro, Stefanie D. Roman.

**Supervision:** Daniel J. Becker, Stefanie D. Roman.

**Validation:** Daniel J. Becker, Kenneth Csehak, Alexander M. Barbaro, Stefanie D. Roman.

**Visualization:** Daniel J. Becker, Kenneth Csehak, Alexander M. Barbaro, Stefanie D. Roman.

**Writing – original draft:** Daniel J. Becker, Kenneth Csehak, Alexander M. Barbaro, Stefanie D. Roman.

**Writing – review & editing:** Daniel J. Becker, Kenneth Csehak, Alexander M. Barbaro, Stefanie D. Roman, Scott Sherman, Sahnah Lim.

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
