## [Decision Letter · Decision Letter 0]

5 Apr 2023

PONE-D-23-02430Experiences of researchers in the Veterans Health Administration during the COVID-19 pandemicPLOS ONE

Dear Dr. Csehak,

Thank you for submitting your manuscript to PLOS ONE. After careful consideration, we feel that it has merit but does not fully meet PLOS ONE’s publication criteria as it currently stands. Therefore, we invite you to submit a revised version of the manuscript that addresses the points raised during the review process.

ACADEMIC EDITOR:

Please ensure that you have submitted a justification for the chosen sample size.

Please ensure that you included the measures taken that the low response rate did not cause any bias

We look forward to receiving your revised manuscript.

We look forward to receiving your revised manuscript.

Kind regards,

Abubakr Abdelraouf Alfadl, Ph.D.

Academic Editor

PLOS ONE

Journal Requirements:

3. We note that Figure 2 in your submission contain copyrighted images. All PLOS content is published under the Creative Commons Attribution License (CC BY 4.0), which means that the manuscript, images, and Supporting Information files will be freely available online, and any third party is permitted to access, download, copy, distribute, and use these materials in any way, even commercially, with proper attribution. For more information, see our copyright guidelines: http://journals.plos.org/plosone/s/licenses-and-copyright.

b.If you are unable to obtain permission from the original copyright holder to publish these figures under the CC BY 4.0 license or if the copyright holder’s requirements are incompatible with the CC BY 4.0 license, please either i) remove the figure or ii) supply a replacement figure that complies with the CC BY 4.0 license. Please check copyright information on all replacement figures and update the figure caption with source information. If applicable, please specify in the figure caption text when a figure is similar but not identical to the original image and is therefore for illustrative purposes only.

4. Please upload a copy of Supporting Information Figure 1 which you refer to in your text on pages 4 and 6

Reviewers' comments:

Reviewer's Responses to Questions

**Comments to the Author**

1. Is the manuscript technically sound, and do the data support the conclusions?

Reviewer #1: Yes

Reviewer #2: Partly

2. Has the statistical analysis been performed appropriately and rigorously? 

Reviewer #1: N/A

Reviewer #2: No

3. Have the authors made all data underlying the findings in their manuscript fully available?

Reviewer #1: Yes

Reviewer #2: No

4. Is the manuscript presented in an intelligible fashion and written in standard English?

Reviewer #1: No

Reviewer #2: No

5. Review Comments to the Author

Reviewer #1: General:

- The manuscript tackles an important issue and the authors used sound methods to obtain data.

Abstract:

- Concise and informative

Introduction:

- Provides a convenient answer to what is known and what is not known and the implication of the study

Methods:

- Well written with relevant details

Results:

- Too long section (8 pages).

- This could be reduced to 2 pages or even less if the authors highlighted only the main findings and refer the reader

to the table or to the figures.

- In table 1, there is an overlap between age groups e.g. from 20-30 and then from 30-40; where to put a patient

with an age of 30 years?

Discussion:

- Discussion is expected to be one unit with paragraphs; each to cover an objective related information. Rewrite

- Avoid putting with subtitles. Remove subtitle in lines 384, 367, 385

- There are 2 different reference styles in the discussion (line 310, 312 vs 315, 318)

- The authors missed to state the limitations. Low response rate is the main limitation.

Others:

- No title seen for figure 1 and 2.

- Figures are of poor quality; redesign

Reviewer #2: 1. Research title can be more specific to oncology researchers

2. Abstract-conclusion: too general. Please write more specifically for this paper.

3. Abstract-would be better to add keywords

4. Introduction: What is novelty in research, statement of problem is not clear.

5. Materials and methods -Unclear calculation of the sample size, how many clinical researchers, faculty were there in that area, and based on that number, calculate the total size (only 118???). Only 43 respondents? These can be preliminary results or a pilot study for this.

6. Response rate was only 36%, what could be the possible reasons and what initiatives were taken to improve participation.

7. Table heading must be on top not at the bottom. Table 1 age categories are confusing, where age group 30 comes under two categories…same goes to other age categories, please check and fix.

8. Figure 1 is not clear so better to modify in excel

9. Results are way too long and must be summarized and presented in a concise manner.

10. The discussion needs to be more specific, say more about the obtained results and explain them in the context of similar studies.

11. What are the strengths and limitations of this study?

12. The conclusion needs to be more specific.

13. The literature needs to be adequately cited in the discussion.

6. PLOS authors have the option to publish the peer review history of their article (what does this mean?). If published, this will include your full peer review and any attached files.

Reviewer #1: **Yes: **Elfatih M. Malik

Associate professor, Faculty of Medicine, University of Khartoum

Reviewer #2: No

---

## [Author Response · Author response to Decision Letter 0]

20 May 2023

Please see Response to Reviewers Letter

---

## [Decision Letter · Decision Letter 1]

16 Aug 2023

Experiences of oncology researchers in the Veterans Health Administration during the COVID-19 pandemic

PONE-D-23-02430R1

Dear Dr. Kenneth Csehak,

We’re pleased to inform you that your manuscript has been judged scientifically suitable for publication and will be formally accepted for publication once it meets all outstanding technical requirements.

Kind regards,

Abubakr Abdelraouf Alfadl, Ph.D.

Academic Editor

PLOS ONE

Additional Editor Comments (optional):

Reviewers' comments:

Reviewer's Responses to Questions

**Comments to the Author**

1. If the authors have adequately addressed your comments raised in a previous round of review and you feel that this manuscript is now acceptable for publication, you may indicate that here to bypass the “Comments to the Author” section, enter your conflict of interest statement in the “Confidential to Editor” section, and submit your "Accept" recommendation.

Reviewer #3: All comments have been addressed

2. Is the manuscript technically sound, and do the data support the conclusions?

Reviewer #3: Yes

3. Has the statistical analysis been performed appropriately and rigorously? 

Reviewer #3: Yes

4. Have the authors made all data underlying the findings in their manuscript fully available?

Reviewer #3: Yes

5. Is the manuscript presented in an intelligible fashion and written in standard English?

Reviewer #3: Yes

6. Review Comments to the Author

Reviewer #3: All of the comments have been handled by the authors, and the work may now be considered for publication.

7. PLOS authors have the option to publish the peer review history of their article (what does this mean?). If published, this will include your full peer review and any attached files.

Reviewer #3: **Yes: **Sairah Hafeez Kamran

---

## [Editor Report · Acceptance letter]

18 Aug 2023

PONE-D-23-02430R1 

Experiences of oncology researchers in the Veterans Health Administration during the COVID-19 pandemic 

Dear Dr. Csehak:

I'm pleased to inform you that your manuscript has been deemed suitable for publication in PLOS ONE. Congratulations! Your manuscript is now with our production department. 

Kind regards, 

on behalf of

Dr. Abubakr Abdelraouf Alfadl 

Academic Editor

PLOS ONE